



# Extratropical cyclones over the North Atlantic and Western Europe during the Last Glacial Maximum and implications for proxy interpretation

Joaquim G. Pinto[1] and Patrick Ludwig[1]

[1]Institute of Meteorology and Climate Research, Karlsruhe Institute of Technology, 76131 Karlsruhe, Germany

*Correspondence to*: Joaquim G. Pinto (joaquim.pinto@kit.edu)

**Abstract.** Extratropical cyclones are a dominant feature of the mid-latitudes, as their passage is associated with strong winds, precipitation, and temperature changes. The statistics and characteristics of extratropical cyclones over the North Atlantic region exhibit some fundamental differences between Pre-Industrial (PI) and Last Glacial Maximum (LGM) climate

conditions. Here, the *statistics* are analysed based on results of a tracking algorithm applied to global PI and LGM climate simulations. During the LGM, both the number and the intensity of detected cyclones was higher compared to PI. In particular, increased cyclone track activity is detected close to the Laurentide ice sheet and over central Europe. To determine changes in cyclone *characteristics*, the top 30 extreme storm events for PI and LGM have been simulated with a regional climate model and high resolution (12.5 km grid spacing) over the eastern North Atlantic and Western Europe. Results show that LGM

extreme cyclones were characterised by weaker precipitation, enhanced frontal temperature gradients, and stronger wind speeds than PI analogues. These results are in line with the view of a colder and drier Europe, characterised by little vegetation and affected by frequent dust storms, leading to reallocation and build-up of thick loess deposits in Europe.

## 1. Introduction

The day-to-day weather conditions in the mid-latitudes are strongly affected by the passage of extratropical cyclones, which

is typically associated with precipitation, strong winds, and changes in temperature and cloudiness. Cyclones also play a major role in the water cycle and the redistribution of momentum and energy in the climate system (Hoskins and Valdez, 1990: Chang et al., 2002). The assessment of cyclone activity, notably to analyse their paths, characteristics and impacts are thus key to determine both the day-to-day weather conditions, the regional mean climate and its variability on multiple time-scales. In fact, there is a wide range of literature analysing case studies of extreme cyclones (e.g., Wernli et al., 2002; Ludwig et al.,

2015), the mean cyclone activity in the mid-latitudes in the recent past (e.g., Hoskins and Hodges, 2002; Ulbrich et al., 2009) and possible changes under future climate conditions (e.g., Bengtsson et al., 2009; Ulbrich et al., 2009). On the other hand, studies analysing the structural characteristics of extratropical storms from a climatological perspective are less frequent (e.g. Catto et al., 2010; Rudeva and Gulev, 2011; Dacre et al., 2012; Hewson and Neu, 2015; Sinclair et al., 2019). While some





general concepts are available on how warmer climate conditions will affect the intensity and structure of cyclones, there are

still several open questions, particularly regarding how dominant the increased latent heating may become compared to other physical processes like low-level and upper-level baroclinicity (see Catto et al., 2019; their Figure 2).

The availability of studies addressing the characteristics of cyclone activity outside of the period extending from the mid-19th Century to the end of the 21st-Century decreases sharply. Raible et al (2018) analysed variations of cyclone statistics in a very long simulation with a fully coupled earth system model from 850 to 2100 CE. While they identified variations on multiple

time-scales, they found no evidence for an external forcing imprint before 1850. Moreover, Pfahl et al. (2015) analysed cyclone activity in idealised aquaplanet simulations covering a wide range of possible climate conditions (from 270K to 316K global mean temperatures). While the structure of the majority of the cyclones reveals only small changes on average, stronger differences were identified for intense cyclones. For example, cross-front temperature differences are expected to be higher (lower) for considerably colder (warmer) climates (Pfahl et al., 2015, their Figure 10), whereas the associated precipitation is

expected to decrease (increase; their Figure 12) due to the strong limiting effect of temperature on the atmospheric moisture content.

One important shortcoming hampering more studies analysing non-recent or non-21st century climate conditions is the availability of climate model data output with sufficient spatial and temporal resolution to enable an adequate identification, tracking and characterisation of the cyclones. For example, model data from the PMIP3 project (Braconnot et al., 2012) are

only archived 6-hourly for short (30 years) time slices, and typically at a low resolution (approximately 200-300 km). Most pre-20th Century studies consider aggregated measures of cyclone (or synoptic) activity (e.g., Kageyama et al., 1999; Laine et al., 2009; Hofer et al., 2012; Ludwig et al, 2016), thus not enabling a detailed comparison to regional temperature and precipitation variability. One period of particular interest is the Last Glacial Maximum (LGM; Clark et al., 2009), when the European climate was characterised by colder and mostly drier conditions (Bartlein et al., 2011; Annan and Hargreaves, 2013;

Újvári et al., 2017; Cleator et al., 2019). Large parts of Northern Europe were covered by permanent ice sheets and surrounded by polar-desert conditions (Ray and Adams, 2001). Western, Central and Eastern Europe were largely characterised by open shrublands and grasslands (steppe-tundra), while in Southern Europe steppe with embedded forest (forest steppe) dominated (Ray and Adams, 2001). Under these conditions, dust storms triggered by strong winds must have been common in Europe, as documented by the major loess deposits found primarily around 50°N over Western and Central Europe and over large parts

of Eastern Europe (Antoine et al., 2009; 2013; Sima et al., 2013; Újvári et al., 2017).

Under the influence of the continental ice sheets and extended sea ice, the PMIP3 GCMs show enhanced meridional temperature gradients, leading to a North Atlantic polar jet which was located further south, more intense and less variable than under current climate conditions (Löfverström et al, 2014; Merz et al., 2015; Wang et al. 2018). These differences have been related with more dominant cyclonic Rossby wave breaking near Greenland (Riviére et al., 2010), stationary wave

reflection (Löfverström et al., 2016) and to enhanced meridional eddy momentum flux convergence over the North Atlantic (Wang et al., 2018). Accordingly, the North Atlantic storm track was more intense and shifted southwards compared to today's



climate (Hofer et al., 2012; Luetscher et al., 2015; Ludwig et al., 2016). The PMIP3 models show indications that while Europe was largely drier than today, this is not the case for some regions, notably for Iberia (e.g., Hofer et al., 2012; Beghin et al., 2016, Ludwig et al., 2016). However, wetter conditions over Iberia are not in line with (most of) the proxy data, which themselves are often associated with considerable uncertainties (e.g., Bartlein et al., 2011; Moreno et al., 2014, Cleator et al., 2019). Nevertheless, the substantial misrepresentation of the regional climate for LGM in PMIP3 models compared to proxies is regarded as a general issue (Harrison et al., 2015). In some cases, such caveats can be partially traced back to shortcomings of the GCMs and/or their boundary conditions. For example, Ludwig et al (2017) provided evidence that the consideration more realistic boundary conditions, particularly North Atlantic SSTs, land use types and vegetation cover, can help derive representations of the LGM climate at the regional scale that are in better agreement with the proxies in terms of temperature, precipitation and the permafrost margin. Still, further studies are needed, notably at the regional scale (e.g. Ludwig et al., 2018), in order to further our confidence in the modelling capabilities and our understanding of the paleoclimate conditions for Europe in key periods like the LGM (Harrison et al., 2015; 2016; Ludwig et al., 2019).

The present work aims to advance our understanding of the LGM climate over the North Atlantic and Europe through a more detailed analysis of the cyclonic activity and its associated impacts, notably in terms of precipitation, temperature and wind speed. The LGM cyclones are first identified and tracked on a simulation with the coupled MPI-ESM-P model, for which data with high temporal resolution was archived. Secondly, a sub-sample of extreme cyclones is downscaled with a regional climate model to analyse possible changes in LGM cyclone characteristics compared to its modern counterparts with high spatial and temporal resolution. The identified characteristics of LGM extreme cyclones are discussed in terms of the available proxies for LGM Climate across Western Europe. Additionally to the precipitation and temperature, the importance of the dominant land cover conditions and the frequent occurrence of dust storms is evaluated. The final section presents the summary and main conclusions.

## 2. Data and Methods

The starting point of our analysis are simulations within the scope of the third phase of the Paleoclimate Modeling Intercomparison Project (PMIP3) (Braconnot et al., 2012) (http://pmip3.lsce.ipsl.fr/). The simulations were performed according to the PMIP3 21 ka experimental design, which includes the lower sea level and blended ice sheet data (Peltier et al., 2015, Lambeck and Chappell, 2001; Lambeck et al., 2002; Tarasov and Peltier, 2002, 2003), orbital parameters and adapted greenhouse gas concentrations (see Table 1). From the available GCMs, we have selected the MPI-ESM-P (Stevens et al., 2013; Jungclaus et al., 2013) constant forcing simulations, for which two 30-year time slices with six hourly output data are available for Pre-Industrial (PI) and LGM conditions. Ludwig et al. (2016) have recently analysed a small ensemble of PMIP3 models in terms of large-scale circulation for Europe, jet stream synoptic activity, precipitation and temperature for LGM. While the MPI-ESM-P model has a slightly different jet structure than some of the other PMIP3 models in terms of the



differences between PI and LGM conditions (cf. Ludwig et al., 2016; their Figure 2), which also impacts for example the storm track and precipitation (their Figures 4, 6), its main characteristics are generally close to the ensemble average.

Individual extratropical cyclones over the North Atlantic and Europe are identified and tracked based on six hourly mean sea level pressure data with a widely used automatic tracking algorithm (Murray and Simmonds, 1991; Pinto et al. 2005). The resulting cyclone statistics provide information on the lifetime of each identified cyclone, thus enabling the computation of mean cyclone statistics like track density, mean maximum intensity, cyclogenesis, cyclolysis, propagation speed and deepening rates. Cyclone statistics obtained with this method compare well with other methodologies (e.g., Neu et al., 2013; Hewson and

Neu, 2015). Following Pinto et al. (2009), cyclones are selected based on the following conditions: (a) cyclone lifetime of at least 24 h hours, (b) a minimum core MSLP value below 1000 hPa (3) a maximum vorticity (approximated by the Laplacian of MSLP) value above 0.6 hPa deg. lat.$^{-2}$, and (d) a maximum deepening rate of 0.3 hPa deg. lat.$^{-2}$s$^{-1}$ is achieved at least once during their lifetime. The method is here applied to the MPI-ESM-P data for the extended (ONDJFM) winter season. In order to analyse the characteristics of the most extreme cyclones affecting Europe in more detail, the most intense 30 cyclones (TOP

30) for the PI and LGM periods are selected based on their peak intensity in terms of vorticity and their passage within a pre-defined box over the eastern North Atlantic (Figure 1, dashed box). The selection of the box enables the creation of a cyclone ensemble that impacts Western Europe and permits a comparison with terrestrial proxy data e.g. for precipitation and dust.

The Weather Research and Forecast (WRF) model (Skamarok et al., 2008) is used in its version 3.9.1.1 to simulate theses cyclones with a final grid spacing of 12.5 km (including 35 vertical layers up to 30 hPa). To achieve a grid spacing of 12.5

km, a 2-step nesting approach is necessary. Cyclones were initially simulated on a 50 km grid forced by MPI-ESM-P data as initial and boundary conditions with an update frequency of six hours. The final 12.5 km grid spacing is achieved by a second nesting step within the WRF model. An overview of the used physical parametrisations is given in Table 2. For the calculation of wind gusts, a gust parametrisation based on 10 m wind speed and friction velocity (Schulz and Heise, 2003; Schulz, 2008) has been implemented into the WRF model. This gust parametrization shows an overall good agreement with observed wind

gusts, particularly over flat terrain (Born et al. 2012). For the WRF simulations, global PI and LGM boundary conditions were adapted considering specifications of the PMIP3 protocol (Braconnot et al., 2012, see also Ludwig et al. 2017). These changes encompass orbital parameters, trace gases (see Table 1), the consideration of ice sheets (extent and height), an associated lowering of the sea level and adaption of land use cover (CLIMAP Project Members, 1984).

WRF cyclone tracks and intensities were identified manually based on relative vorticity at 850 hPa. For comparison of the PI

and LGM cyclone characteristics, and following the methodology from Catto et al. (2010; their Figure 3), each track was rotated so that the cyclones were equally moving in west-east direction, enabling the generation of composites for different atmospheric variables (cf. also Dacre et al., 2012). Composites have been created for peak intensity (0) and 6, 12,18 and 24 hours before peak intensity, and 6 and 12 hours afterwards. For the sake of succinctness, we will primarily discuss the time frames i) 12 hours before peak intensity and ii) peak intensity. The here analysed target variables, based on the 12.5 km WRF



simulations, include mean sea level pressure, precipitation, column integrated water vapour, equivalent-potential temperature 850 hPa, 925 hPa winds, near-surface wind gusts.

## 3. Northern Hemisphere Cyclone Statistics for PI and LGM conditions

In this section, we analyse the general characteristics of cyclones over the North Atlantic and Europe under LGM conditions and compare them to PI climate conditions. Figure 2 shows the cyclone track density for the extended winter for PI and LGM

climate conditions. In spite of the lower spatial resolution of MPI-ESM-P, the cyclone track density for the PI is close to cyclone statistics obtained with Reanalysis datasets and CMIP GCMs for recent climate conditions (cp. Pinto et al., 2007; their Figure 1). Still, some regional shortcomings are identified, notably the limited cyclone activity over the Mediterranean basin. The North Atlantic storm track shows a clear tilt towards Northern Europe and the Arctic Ocean for PI, and its location and orientation are closely related with the eddy-driven jet stream (black contours in Fig. 2a) and the associated upper-air

baroclinicity (Hoskins and Valdes, 1990; Pinto et al., 2009).

The North Atlantic storm track looks quite different under LGM conditions: the cyclone track density is strongly enhanced over the North Atlantic and more constraint along a corridor close to the ice edge (Fig. 2b). Close to Europe, a bifurcation is found, and cyclones are either deflected northward along the border of the Scandinavian ice sheet or south-eastward towards Central Europe and the Mediterranean (Fig. 2c). In accordance, the eddy-driven jet is stronger under LGM conditions in the

MPI-ESM-P, thus establishing more favourable conditions for the occurrence of intense storms affecting Western and Central Europe. For the North Atlantic ($70°W – 0°$, $35°N – 70°N$), the total number of cyclones for the analysed 30-year period is about 26% larger for LGM than for PI conditions (12.071 vs. 9.541 individual cyclone counts).

Other properties of cyclone activity are depicted in Figure 3 for LGM in terms of absolute values (left column) and their differences to PI (right column). Cyclogenesis is dominant along the North American east coast for LGM (Fig. 3a), and much

stronger than in PI (Fig. 3b), which is in line with a much stronger upper level jet stream during glacial conditions. Moreover, cyclogenesis is enhanced south of Greenland and over Western and Central Europe. On the other hand, cyclolysis is enhanced along the borders of the Greenland and Scandinavian ice sheet (Fig. 3c), in strong deviation to the PI conditions (Fig. 3d). Mean maximum cyclone intensity is typically attained in a region extending from Newfoundland to Iceland and the British Isles, with a secondary maximum over Eastern Europe (Fig. 3e). Compared to the PI cyclones, the LGM cyclones reveal

stronger intensities, particularly in an area extending from the south of Greenland to the British Isles, and over most of continental Europe. On the other hand, cyclone intensity close to the North American East coast is considerably lower (Fig. 3f). Deepening rates are particularly stronger for LGM cyclones over the central north Atlantic, and filling rates close to the ice edge / ice shields (Fig. 3g). This is in strong contrast with PI cyclones (Fig. 3h), which comparatively have stronger deepening rates further upstream, and weaker deepening and filling rates. These results point towards different typical





development of LGM cyclones compared to their PI counterparts, which occurs either more zonally at lower latitudes towards Central Europe or further downstream closer to the ice edge towards the Arctic.

Figure 4 displays the relative frequency distribution of the cyclone intensity over the North Atlantic area (70°W – 0°, 35°N – 70°N), revealing that LGM cyclones are on average more intense than their PI counterparts. In particular, the number of cyclones exceeding 3 hPa deg. lat.$^{-2}$s$^{-1}$ is twice as large for LGM than for PI. In fact, a small number (18) of LGM cyclones

attain intensities exceeding the range identified for PI cyclones. The statistics for the region close to Europe (box) are similar (not shown). All these results document a shift towards stronger intensities for LGM cyclones, both in terms of average numbers and extreme values.

## 4. Characteristics of Extreme Cyclones over the Eastern North Atlantic

To analyse the characteristics of cyclones for the LGM, a subset of 30 extreme cyclones passing over the Eastern North Atlantic

is selected for both the PI and LGM periods based on the highest values for vorticity (Laplacian of MSLP) within the selected box. The trajectories of the selected MPI-ESM cyclones are depicted in Figure 5 and key information are given in Table 3. Two important facts are clearly identifiable: LGM extreme cyclone trajectories are more zonally orientated and constrained to a narrower corridor (particularly until 15°W) than their PI counterparts, and they achieve higher vorticity (Laplacian of MSLP) values during lifetime (Tab. 3).

For each of these cyclones, WRF simulations down to 12.5 km grid-spacing are performed along the most intense segment of their lifetimes, thus gaining 3D data to analyse the cyclones e.g. in terms of the evolution of their structure, air masses, winds and precipitation. Care was taken that storms agree between the original MPI-ESM-P data and WRF cyclone tracks. Generally, the obtained tracks based on the WRF simulations reveal lower core pressures and higher vorticity than their low resolutions counterparts do. Figure 6 shows a comparison between high and low resolution data for a selected cyclone. The considered

cyclone tracks (MPI-ESM LGM cyclone #24) show a good superimposition for MPI-ESM, WRF 50 km and WRF 12.5 km and a good agreement of the position of maximum intensity with a slightly further south-eastward location for WRF 50 km (Fig. 6a). While the time series, centred at peak intensity, for the development of core pressure show a similar strong pressure drop for all resolutions, the relative vorticity exhibits much stronger values for the high resolution simulation (WRF 12.5 km) in comparison with the coarser realisations. Note that the WRF simulations do not cover the whole trajectory of the cyclones,

and thus the time series are shorter. Figure 6c and 6d depict the corresponding precipitation patterns at peak intensity, in this case revealing more small scale structures and higher precipitation values for the 12.5 km WRF simulation.

Based on the top 30 cyclones for PI and LGM and using the composite methodology described in section 2, average cyclone characteristics are investigated for the intensification phase, focussing on the time frames (i) 12 hours before peak intensity and (ii) peak intensity. Figure 7 displays the MSLP fields for PI and LGM for peak intensity; the corresponding panels for 12

hours before peak intensity can be found in Fig. S1. The anomalies compared to the mean MSLP are displayed in colours, in





order to analyse MSLP gradients (Fig. 7 a,b; S1 a,b). The core MSLP of the LGM has higher values compared to PI (LGM: 982.7 hPa, PI: 975.2 hPa). This can be explained by the lower sea levels (~120m) for LGM, causing a difference of global mean sea level pressure of about ~13 hPa (PI: 1010 hPa, LGM: 1023 hPa). Taken this into account, the LGM cyclones reach deeper MSLP values compared to the global average MSLP, being consistent with the stronger deepening rates for LGM cyclones identified by the tracking algorithm (Fig.3 g,h). Additionally, the closer isobars south of the cyclone core indicate stronger pressure gradients for the LGM cyclones, which is supported by the LGM – PI differences (Fig. 7c). This is particularly the case on the expected location of the frontal areas. Fig. 7 (d,e) and S1(d,e) displays the anomalies of potential temperature at 850 hPa. Results show that the cross-frontal gradients are particularly intense across the warm front and that the whole development is apparently faster for LGM conditions, indicating a faster occlusion (Fig. 7 f). On the other hand, the total water content is much higher under PI conditions, primarily due to the effect of the higher environmental temperatures (Fig. 7 g,h; S1 g,h) with differences in the warm sector reaching up to 10 mm and 6-8 mm close to the cyclone core at peak intensity (Fig. 7i).

The strong difference in available water content again leads to a large difference in terms of accumulated precipitation, which is clearly larger for the PI composites (Fig. 8 a,b; Fig. S2 a,b). While some of the smaller deviations can be potentially attributed to slightly different developments, the total precipitation is considerably lower for LGM extremely cyclones (up to 1.7 mm h$^{-1}$) particularly in the area of peak precipitation close to the cyclone centre (Fig. 8c). On the other hand, wind speed at 925 hPa is strongly increased for LGM cyclones (Fig. 8 d,e; Fig. S2 d,e), where wind speeds over large areas south of the cyclone exceed their PI counterparts by 10-12 m/s, particularly for 12 hours before peak intensity (Fig. 8 f; S2f). Strong differences are also revealed for near-surface wind gusts (Fig. 8 g-i; S2g-i). In this case, the wind gusts are particularly enhanced along the expected location of the cold front at peak intensity, with deviations exceeding 5 m/s. The above described patterns remain true for the cyclone characteristics 12 hours after peak intensity (Figs S3, S4). For example, stronger wind gusts remain dominant in the area south of the cyclone core (Fig. S4i).

In summary, LGM cyclones display steeper MSLP gradients (in agreement with higher intensity in terms of circulation), larger temperature gradients between the air masses, weaker precipitation and stronger wind gusts than their PI counterparts do.

## 5. Discussion with available proxy-based climate reconstructions

The above analysis shows that extreme cyclones under LGM conditions were typically more intense than PI extreme cyclones. Likewise, they were associated with larger frontal temperature gradients, stronger winds and reduced precipitation. In this section, we analyse in how far these cyclone characteristics can help to explain the occurring climate in Western and Central Europe under LGM conditions and the frequent occurrence of dust storms in that area.



According to the available proxy records, Western and Central Europe were colder and largely drier under LGM conditions (Bartlein et al., 2011; Annan and Hargreaves, 2013; Cleator et al., 2019), and was to a large extent covered by open shrublands and grasslands (Ray and Adams, 2001). The colder and generally drier conditions are reproduced by the PMIP3 GCMs, but considerable differences in detail have been identified compared to proxy-based climate reconstructions (Beghin et al., 2016; Ludwig et al., 2016; Cleator et al., 2019). These can be both attributed to shortcomings of the GCMs and/or uncertainties in
the reconstructions, which are often not well constrained (Bartlein et al., 2011; Ludwig et al., 2019; Cleator et al., 2019). For example, most PMIP3 GCMs show enhanced precipitation over the Iberian Peninsula compared to PI climate (Beghin et al., 2016; Ludwig et al., 2016), which disagrees with the proxy data (e.g. Bartlein et al., 2011). Later, Ludwig et al (2017, 2018) show that the wet bias over Iberia and Western Europe can strongly be reduced by considering more realistic boundary conditions (particularly in terms of sea surface temperatures, land use and vegetation cover) in high-resolution RCM
simulations.

The present results enable a more detailed evaluation of the above hypotheses and interpretations. Under current climate conditions, precipitation in the mid-latitudes is largely associated with the passage of extratropical cyclones, where extreme cyclones have a comparatively larger contribution to total/extreme precipitation (Pfahl and Wernli, 2012; Hawcroft et al. 2012). Hawcroft et al. (2016) have provided evidence that GCMs typically have some shortcomings in representing the precipitation
associated with mid-latitude cyclones. It is reasonable to assume that this caveat may be exacerbated at lower resolutions, e.g. those typical of the PMIP3 models. An idealised study by Pfahl et al. (2015) had shown that precipitation increases (decreases) disproportionately in considerably warmer (colder) climate conditions. With the help of the high resolution WRF simulations, we could provide evidence that extreme cyclones under LGM conditions indeed induce considerably less precipitation than their PI counterparts (~22% within a 10° radius around the cyclone centre). Even though this effect could be somewhat
compensated by the moisture advection embedded in the (stronger) westerly large-scale flow, particularly for areas where orographic precipitation dominates (e.g. upward slopes of mountain ranges / glaciers), this lower precipitation associated with extratropical cyclones is consistent with the hypothesis of a drier Western and Central Europe, and thus with the dominant land cover types (polar desert close to the glaciers, forest steppe over Southern Europe and steppe in-between) estimated from proxy data (Ray and Adams, 2001) or statistical reconstruction based on temperature and precipitation (Shao et al., 2018).
The fact that we have selected an area close to the Iberia peninsula for the selections of the cyclones leads to the strong assumption that largely drier conditions must also have been found for Southwestern Europe, in agreement with the proxies (e.g. Bartlein et al., 2011, Moreno et al., 2014).

Mineral dust plays an important role in our climate system (Shao et al., 2011). Dust emissions are typically initiated by wind stress on land surfaces with little to no vegetation cover and easily erodible soils (e.g., Prospero et al., 2002). Such areas were
very common in Europe under LGM conditions (e.g., Ray and Adams, 2001, Ugan and Byers, 2007), when the global dust cycle is estimated to have been more intense than in recent times (Maher et al., 2010). A large number of loess deposits over Western and Central Europe (particularly around 50°N; Antoine et al., 2009) document of a past when dust storms were a



common feature of the European climate. In particular, Antoine et al. (2009) identified cyclic variations in loess deposition between 34-17 ka on several sites in France, Germany and Belgium, with particularly high sedimentation rates, and attributed

these to numerous and intense dust storms in periods with adequate large-scale circulation and reduced precipitation. Furthermore, the high accumulation rates of loess in the middle and lower Danube basin indicate cold, dry and windy conditions during the LGM in south-eastern Europe (Fitzsimmons et al, 2012), consistent with increased storm activity over Central Europe (Fig. 2c). These and other findings document a more intense (global) dust cycle for LGM conditions (e.g. Albani et al., 2016; Újvári et al., 2017; Albani and Mahowald, 2019). The occurrence of dust storms over Western and Central

Europe has been conceptually associated with the passage of intense extratropical cyclones penetrating deep into the European continent (Antoine et al., 2009; their Figure 12). Following on previous studies (e.g. Laine et al., 2009; Hofer et al. 2012; Ludwig et al., 2016), the present results provide evidence for the first time that individual LGM cyclones would be indeed capable of triggering such dust events: their frequent tracks over Western/Central Europe and strong wind speeds could easily trigger dust emissions and transport it over short (for coarse grain) and large (for fine grain material) distances (cp. Shao et al.

2011). As the precipitation associated with these cyclones (which could potentially hamper the triggering of dust storms) is considerably reduced, it should be a less relevant limiting factor under LGM conditions. In addition to the role of the westerlies and embedded cyclones into generating dust storms in Europe, there is evidence that situations with persistent easterlies associated with anticyclonic flow triggered by a strong anticyclone over the Scandinavian ice sheet may have also played a significant role for loess deposition not only over Eastern Europe but also over Central Europe (Újvári et al., 2017;

Schaffernicht et al., 2019).

## 6. Summary and Conclusions

The statistics and characteristics of extratropical cyclones over the North Atlantic and Western Europe were analysed for time-slice experiments for PI and LGM conditions. First, the statistics of the climatologies of PI and LGM cyclones are analysed and compared based on global MPI-ESM-P simulations. Second, the characteristics of extreme LGM cyclones over the Eastern

North Atlantic were analysed in detail based on high-resolution simulations (12.5 km grid-spacing) with the regional climate model WRF and compared to their PI counterparts. The results were discussed with available proxy reconstructions of climate parameters and vegetation types. The main conclusions are as follows:

- The North Atlantic storm track was more intense under LGM conditions, featuring an enhanced number and more intense synoptic systems than under PI conditions. One of the downstream branches brought more often extreme

cyclones towards Western and Central Europe and the Mediterranean area.

- LGM cyclones were more intense due to stronger baroclinicity and apparently less influenced by diabatic processes (lower rainfall, lower temperatures, lower water vapour content). In particular, LGM cyclones benefit from a stronger



and extended jet stream. The development was typically faster, with deepening rates and peak intensities exceeding those from PI cyclones.

- LGM extreme cyclones were characterised by lower precipitation, enhanced frontal temperature gradients, and stronger mean wind speeds and wind gusts than PI analogues.
- These characteristics are in line with the view of a colder and drier Europe, characterised by steppe/tundra land types and affected by frequent dust storms, leading to reallocation and build-up of thick loess deposits.

Given that this study is based on a single GCM, a single tracking method and a single RCM, it should be regarded as a
preliminary analysis as the uncertainties may be considerable. Still, the identified differences between PI and LGM (extreme) cyclones are unequivocal, are consistent with idealised studies, demonstrate the potential of the approach and may become instrumental to facilitate a better interpretation of LGM proxy data. In particular, this study provides new understanding of the relationship between the large scale mean cyclone activity and short term variability on the regional scale and thus may help to reduce numerical interpretative uncertainties (Harrison et al. 2016).

Even though the added value of regional climate models in paleoclimate applications is still controversially discussed (Armstrong et al., 2019), there is a general call for improvements towards a new generation of reliable regional projections (e.g., Harrison et al., 2015; Kageyama et al., 2018). The present and other studies (see Ludwig et al., 2019 for a review) provide clear arguments in favour of an extended used of regional climate models in the scope of paleoclimate studies, as they can play an important role towards a meaningful joint interpretation of proxies and climate model data. The upcoming new paleoclimate
simulations within CMIP6/PMIP4 (Kageyama et al., 2017, 2018), as well as new proxy-based reconstructions of climates (Cleator et al. 2019), will provide novel possibilities to expand our understanding of past climates and to reduce uncertainties on both the numerical and reconstruction branches.

**Data availability:** WRF-Data presented in the paper can be accessed by contacting the authors. The data will be archived at the DKRZ (German Climate Computing Centre). PMIP3 boundary conditions can be obtained at https://pmip3.lsce.ipsl.fr/
(last access: 12 November 2019). Vegetation cover and Landuse data from CLIMAP can be obtained at https://iridl.ldeo.columbia.edu/SOURCES/.CLIMAP/.LGM/ (last access: 12 November 2019).

**Author contributions:** Both authors contributed equally to this work. JGP and PL designed the study and the experiments. PL developed the paleo-specific model adjustments, performed the simulations and prepared the figures. JGP wrote the first draft of the manuscript. JGP and PL contributed with revisions.

**Competing interests**: The authors declare that they have no conflict of interest.



**Acknowledgements:** We thank the German Climate Computing Centre (DKRZ, Hamburg) for providing the MPI-ESM-P data and computing resources within DKRZ Project 965 "Our Way to Europe – Palaeoclimate and Palaeoenvironmental reconstructions". JGP thanks the AXA Research Fund for support, and both authors thank REKLIM (Helmholtz Climate Initiative regional climate change) for partial funding. Thanks to Alexander Reinbold for his contribution to a preliminary
analysis, and Sven Ulbrich for help with the cyclone statistics. The authors also thank the PALEOLINK project by the PAGES 2kNetwork coordinators. We acknowledge support by Deutsche Forschungsgemeinschaft (DFG) and Open Access Publishing Fund of Karlsruhe Institute of Technology.

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





**Tables**

**Table 1.** Boundary conditions adapted in the WRF simulations for PI and LGM based on the PMIP3 protocol.

|  | $CO_2$ | $N_2O$ | $CH_4$ | Eccentricity | Obliquity | Angular Precession |
|---|---|---|---|---|---|---|
| PI | 280 ppm | 270 ppb | 760 ppb | 0.01672 | 23.446 ° | 102.04 |
| LGM | 185 ppm | 200 ppb | 350 ppb | 0.01899 | 22.949 ° | 114.42 |


**Table 2.** Physical parametrization schemes used in the regional model simulations (same parametrizations used for 50 km and

12.5 km domain).

|  | microphysics | cumulus scheme | PBL-scheme | Radiation scheme | surface |
|---|---|---|---|---|---|
| WRF-namelist option | **95** (Eta (Ferrier)) | **6** (Tiedke) | **2** (Mellor–Yamada–Janjic) | SW/LW: **4** (RRTMG) | **2** (Noah–LSM) |
| Reference | Ferrier et al. (1995) | Zhang et al. (2011) | Janjic (1994) | Iacono et al. (2008) | Tewari et al. (2004) |



**Table 3.** Overview of time and location of maximum intensity (defined by the maximum of the Laplacian of mslp) of the TOP

30 MPI-ESM-P cyclones for PI and LGM conditions inside the box (Figure 1).

| | PI | | | | | LGM | | | | |
|---|---|---|---|---|---|---|---|---|---|---|
| | Date | Time | Lapl P | °Lat | °Lon | Date | Time | Lapl P | °Lat | °Lon |
| 1 | 29930107 | 18 | 3.162 | 51.54 | 342.98 | 19360211 | 00 | 3.535 | 52.33 | 338.93 |
| 2 | 29960104 | 12 | 3.158 | 51.96 | 343.35 | 19311012 | 18 | 3.430 | 47.35 | 335.87 |
| 3 | 29901123 | 18 | 2.858 | 44.31 | 340.09 | 19361122 | 00 | 3.220 | 46.53 | 341.67 |
| 4 | 29941108 | 18 | 2.811 | 41.67 | 339.00 | 19421024 | 06 | 3.202 | 52.34 | 345.53 |
| 5 | 29830317 | 06 | 2.786 | 45.87 | 335.95 | 19220117 | 06 | 3.067 | 46.62 | 337.56 |
| 6 | 29980131 | 00 | 2.734 | 49.97 | 336.31 | 19400228 | 12 | 2.965 | 50.20 | 348.23 |
| 7 | 29941115 | 06 | 2.628 | 49.90 | 343.87 | 19411118 | 06 | 2.962 | 49.70 | 343.95 |
| 8 | 29800114 | 18 | 2.615 | 47.33 | 347.16 | 19321113 | 00 | 2.917 | 52.11 | 346.19 |
| 9 | 29980225 | 18 | 2.572 | 49.45 | 335.94 | 19271125 | 06 | 2.911 | 43.70 | 337.33 |
| 10 | 29991024 | 12 | 2.551 | 50.82 | 345.56 | 19211225 | 06 | 2.894 | 45.19 | 345.36 |
| 11 | 29831230 | 06 | 2.541 | 41.97 | 336.09 | 19251112 | 18 | 2.850 | 46.88 | 348.18 |
| 12 | 29851122 | 12 | 2.528 | 48.60 | 349.10 | 19440128 | 00 | 2.823 | 43.52 | 335.73 |
| 13 | 29990207 | 18 | 2.512 | 43.76 | 344.89 | 19400325 | 18 | 2.812 | 49.81 | 335.18 |
| 14 | 29940209 | 12 | 2.511 | 46.31 | 345.74 | 19420113 | 00 | 2.746 | 52.13 | 337.36 |
| 15 | 29891222 | 00 | 2.493 | 51.81 | 342.91 | 19371028 | 00 | 2.728 | 45.43 | 348.59 |
| 16 | 29790224 | 00 | 2.491 | 50.35 | 344.83 | 19230328 | 18 | 2.725 | 50.70 | 349.34 |
| 17 | 30031202 | 06 | 2.489 | 42.45 | 344.77 | 19401105 | 18 | 2.688 | 51.82 | 346.50 |
| 18 | 30010105 | 18 | 2.473 | 48.50 | 339.35 | 19220108 | 06 | 2.679 | 42.31 | 342.38 |
| 19 | 29850119 | 18 | 2.471 | 43.52 | 338.38 | 19230204 | 12 | 2.678 | 48.73 | 342.70 |
| 20 | 29870105 | 12 | 2.466 | 47.53 | 345.43 | 19350103 | 18 | 2.663 | 51.29 | 339.74 |
| 21 | 29961123 | 00 | 2.457 | 46.23 | 337.72 | 19431124 | 18 | 2.644 | 43.34 | 345.30 |
| 22 | 30001223 | 06 | 2.299 | 47.43 | 345.84 | 19351217 | 00 | 2.635 | 52.12 | 342.00 |
| 23 | 29900221 | 18 | 2.298 | 51.33 | 348.96 | 19240223 | 00 | 2.613 | 50.39 | 344.81 |
| 24 | 29811218 | 06 | 2.272 | 47.65 | 345.85 | 19331109 | 06 | 2.611 | 51.09 | 341.01 |
| 25 | 29910118 | 00 | 2.263 | 49.51 | 338.82 | 19381003 | 12 | 2.590 | 51.91 | 336.29 |
| 26 | 29791122 | 12 | 2.252 | 45.25 | 349.74 | 19430211 | 06 | 2.584 | 48.11 | 348.10 |
| 27 | 30040114 | 06 | 2.246 | 44.70 | 340.05 | 19320213 | 06 | 2.507 | 49.92 | 345.95 |
| 28 | 29970210 | 18 | 2.200 | 48.06 | 343.77 | 19270114 | 06 | 2.497 | 47.66 | 341.06 |
| 29 | 30000325 | 00 | 2.167 | 50.88 | 341.08 | 19371204 | 00 | 2.481 | 43.06 | 336.95 |
| 30 | 30031031 | 18 | 2.147 | 46.78 | 347.93 | 19230203 | 00 | 2.478 | 50.41 | 347.84 |





**Figures**

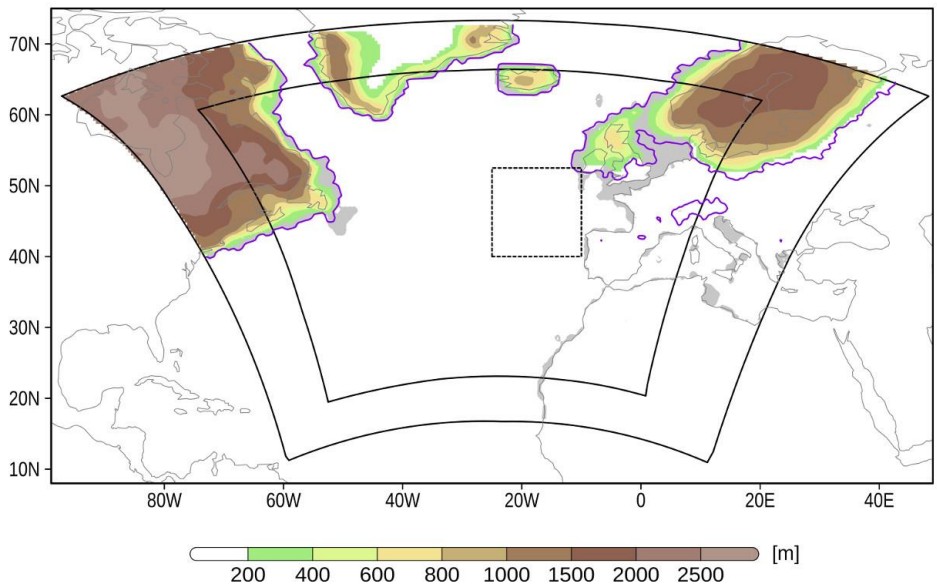

**Figure 1.** WRF model domains (outer solid box: 50 km grid spacing; inner solid box: 12.5 km grid spacing), ice sheet heights
[m] (coloured) and extends (purple line), land sea mask (additional land areas grey) as obtained from PMIP3; target area for
cyclone detection marked by dotted box.



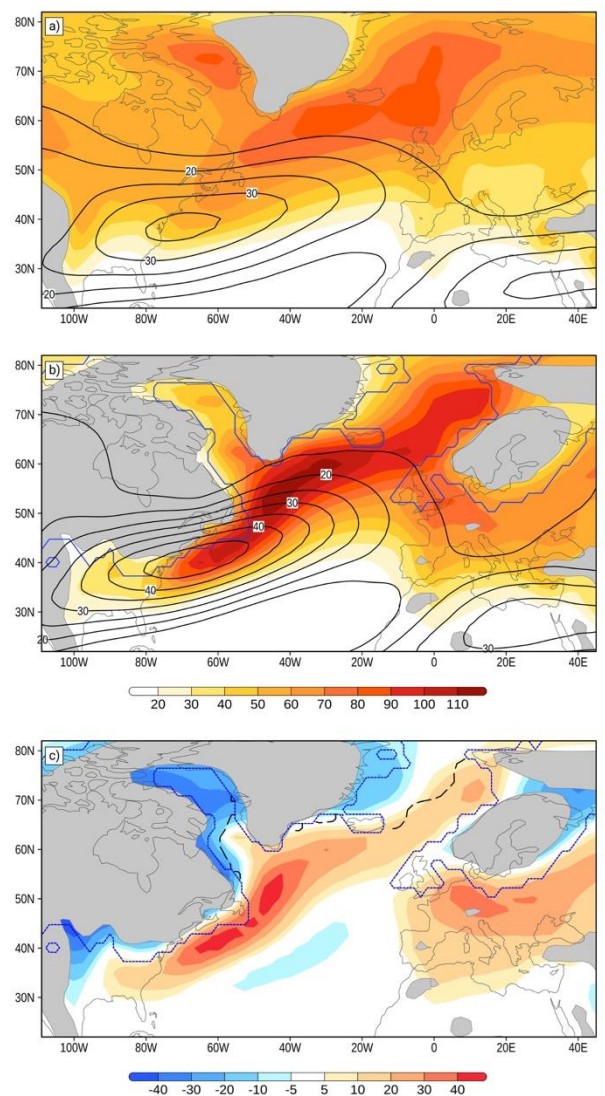

**Figure 2.** Cyclone track density [cyclone days per extended winter per (deg.lat.)$^2$] (coloured) and 300 hPa wind speed [m s$^{-1}$] (contours) based on MPI-ESM-P data for (a) PI, (b) LGM and (c) difference between LGM and PI. Areas with topography higher 1000 m shaded grey, ice sheet margins (b and c) denoted by thin stippled line, long dashed black line in (c) denotes margin of 40% annual sea ice cover.

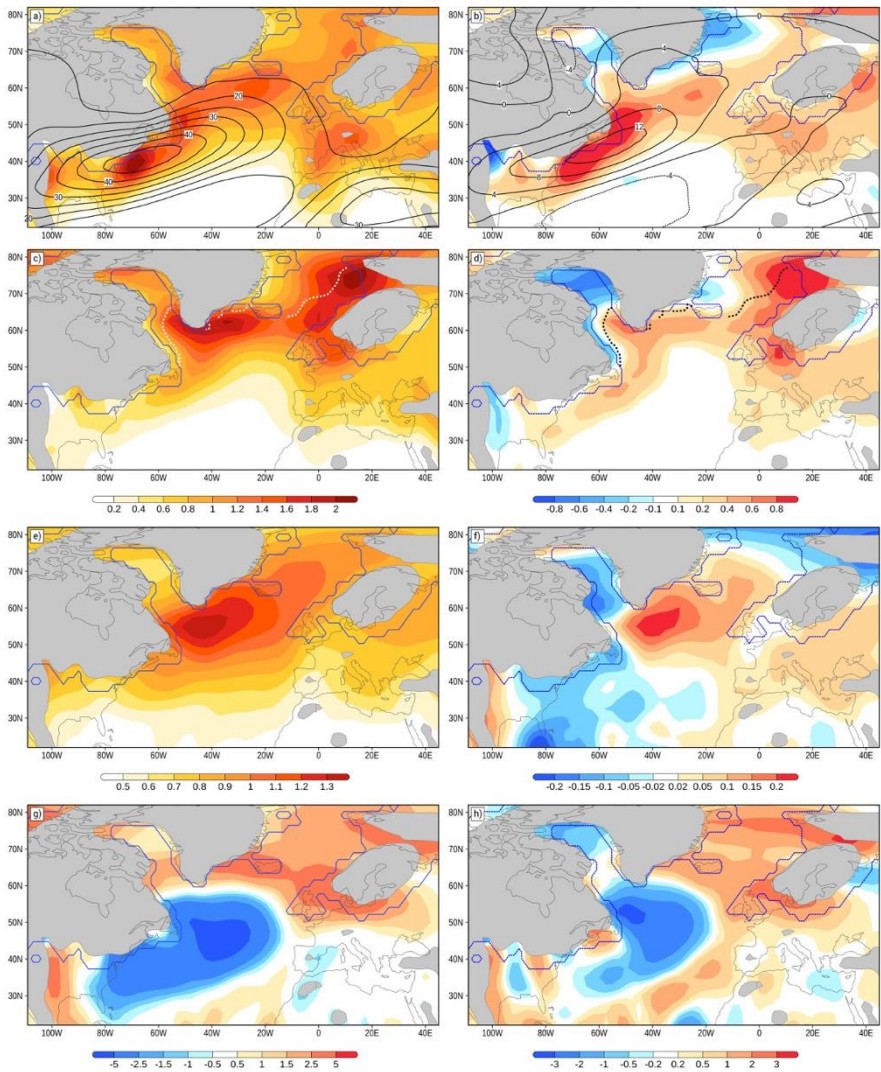

**Figure 3.** Statistical measures obtained from cyclone tracking algorithm for LGM cyclones (left column) and difference to PI cyclones (right column) for (a), (b) cyclogenesis [events per extended winter per (deg.lat.)$^2$] (coloured) and wind speed at 300hPa [m s$^{-1}$] (contours), (c),(d) cyclolysis [events per extended winter per (deg.lat.)$^2$]; (e), (f) mean Δ MSLP [Laplacian of pressure per extended winter per (deg.lat.)$^2$] and (g), (h) deepening rates [hPa h$^{-1}$]. Areas with topography higher 1000 m shaded grey, ice sheet extends marked by blue line. Sea ice margin (>40% annual cover) in (c) and (d) indicated by bold dashed lines.





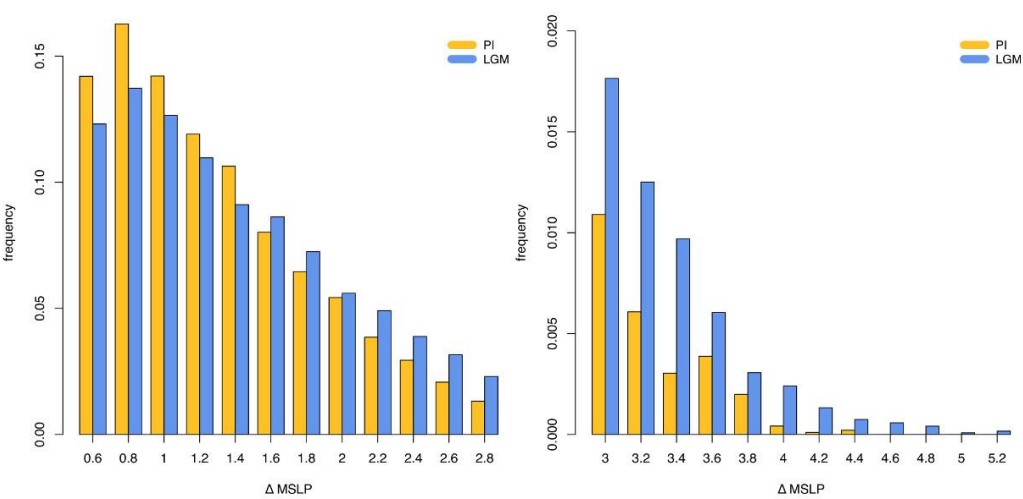

**Figure 4.** Histogram of cyclone intensity (Laplacian ($\Delta$) MSLP) over the North Atlantic (70°W – 0° and 35°N – 70°N). For intense cyclones ($\Delta P \geq 3$), the y-axis is zoomed in.





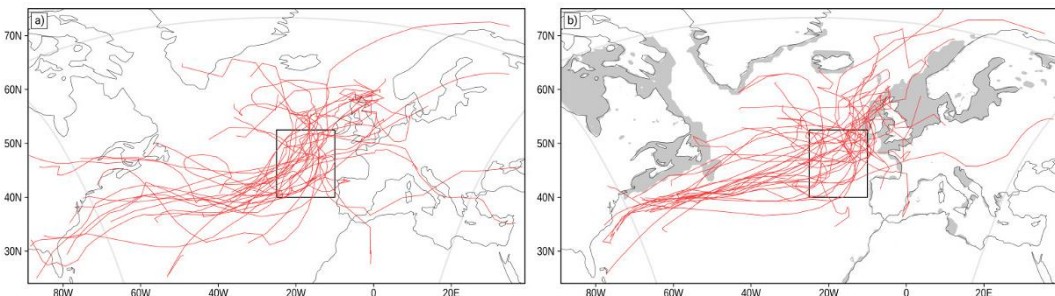

Figure 5. Cyclone tracks of TOP 30 (a) PI and (b) LGM cyclones (MPI-ESM). Black box is region where cyclones need to have maximum intensity to be considered in the composite analysis.


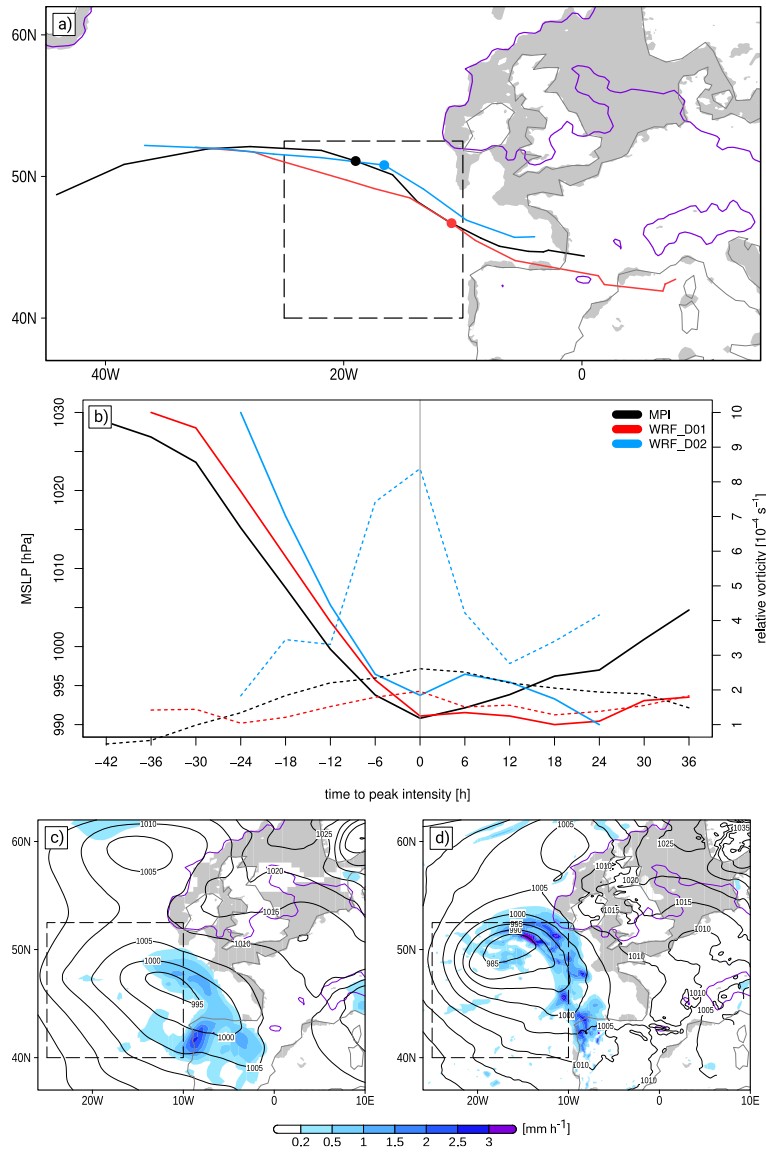

**Figure 6.** Comparison of (a) cyclone tracks (black dotted box shows target area), (b) timeseries of cyclone core pressure and relative vorticity for MPI-ESM (black), WRF 50km (red) and WRF 12.5 km (blue) for LGM cyclone #24. Simulated precipitation rate [mm h⁻¹] at peak intensity for (c) WRF 50km and (d) WRF 12.5km.

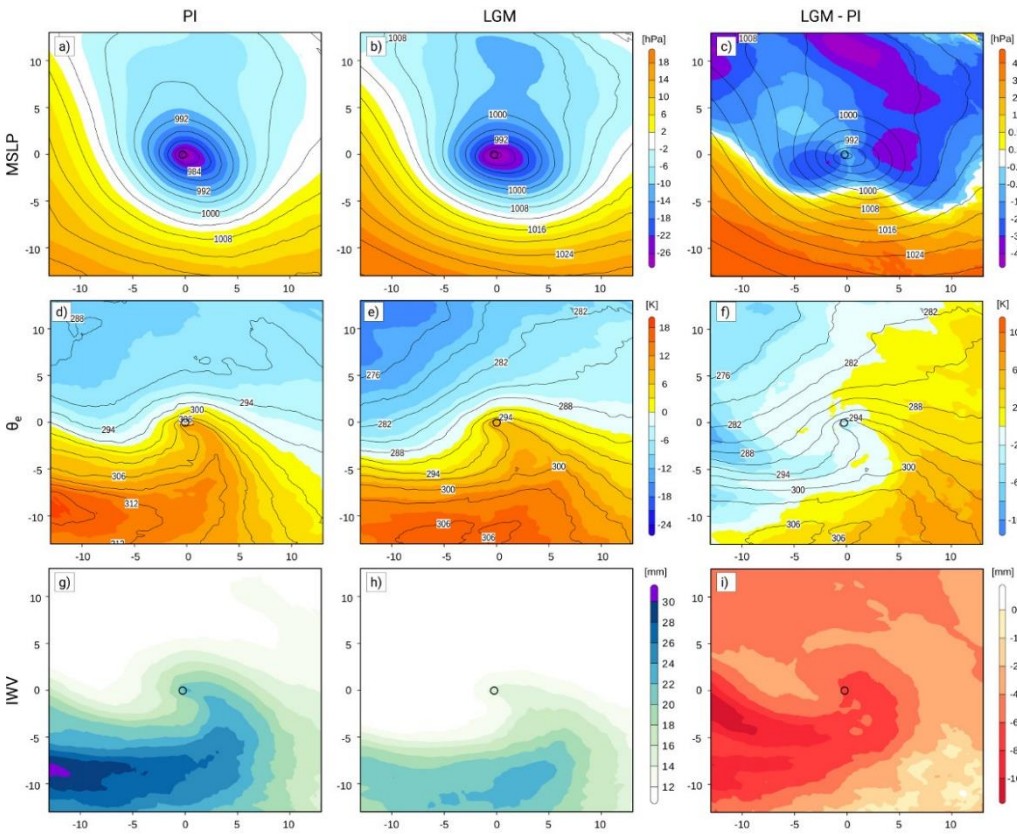

**Figure 7.** Composites of (a - c) mean sea level pressure, (d, e) ThetaE, and (f - g) vertical integrated water vapour (IWV) for PI, LGM and difference LGM – PI at peak intensity as defined by the maximum of the Laplacian of MSLP. (a, b) absolute MSLP values (lines, [hPa]), anomalies [hPa] from field mean (coloured); (c) absolute MSLP values (lines, [hPa]) from LGM, differences LGM – PI in colours; (d, e) absolute ThetaE values (lines, [K]) and anomalies [K] from field mean (coloured); (f, g) absolute IWV values [mm], (h) difference [mm] LGM – PI.

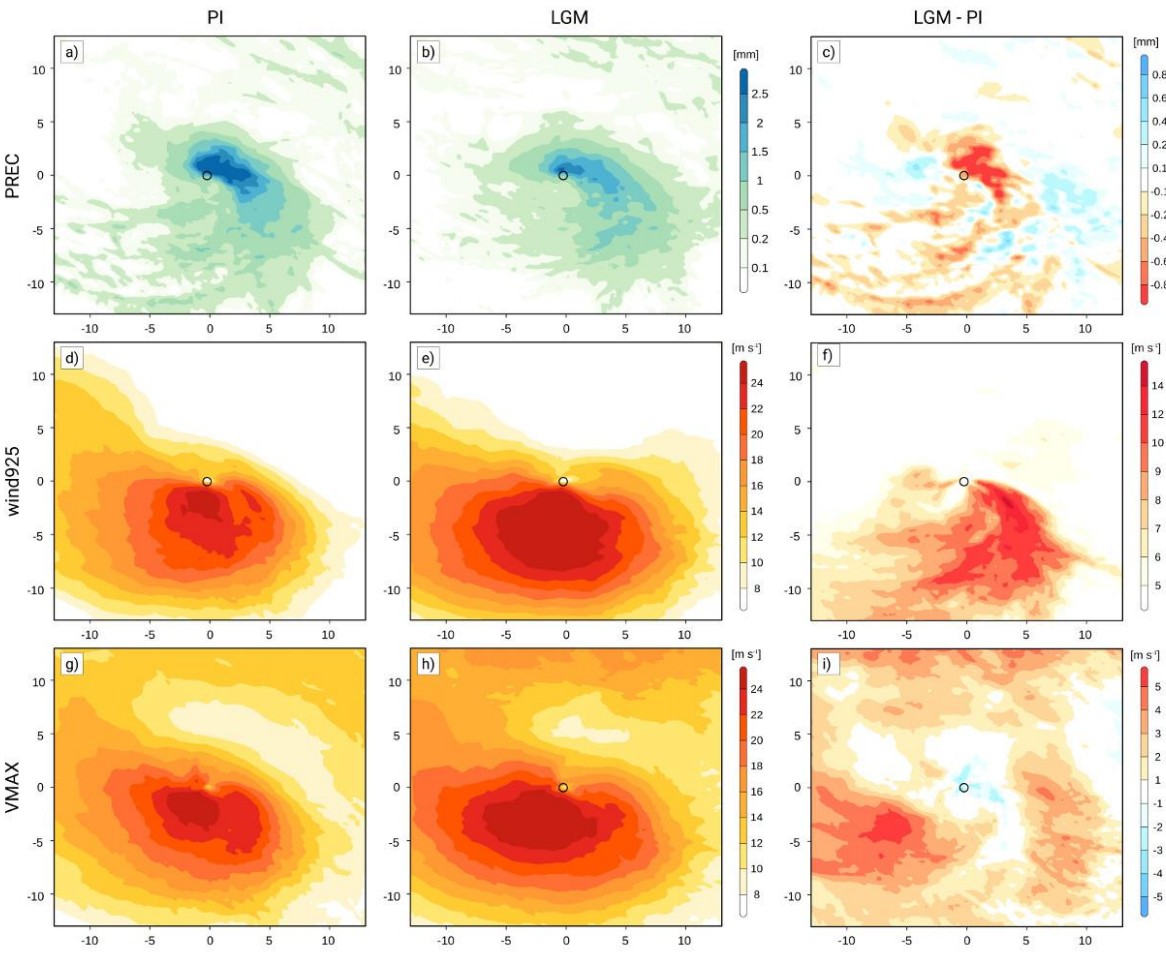

**Figure 8.** As Figure 7 but for hourly precipitation [mm] (a) PI, (b) LGM, (c) LGM – PI, (d - f) wind speed in 925 hPa [m s⁻¹] and (g - i) maximum near surface wind gust [m s⁻¹] at peak intensity.