# Peer review of "Extratropical cyclones over the North Atlantic and Western Europe during the Last Glacial Maximum and implications for proxy interpretation"

_Climate of the Past, 2019_

## Referee Comment (RC1) · Anonymous Referee #1 · 16 Dec 2019

GENERAL COMMENTS

This paper outlines the use of cyclone tracking to identify some key points of the climate and weather over the North Atlantic and Western Europe during that Last Glacial Maximum (LGM). The paper identifies changes in the mean state that may be both conducive to dust emission (e.g. increase in wind speeds) and also those which may hinder it (e.g. more storm activity and associated precipitation). The key point in the authors' analysis is the cyclone tracking and compositing, which provides important information on how the mean climate state is felt at the surface through the extratropical

cyclones (in this case, the 30 most extreme). It is very clear from their results that the strengthening of the thermal gradients in the LGM (relative to the base period) simulations do not cause an increase in precipitation (at the fronts) and that overall there is less precipitation in the extratropical cyclones (for clear physical reasons, which are stated). The high-resolution model data is also useful for reaching their conclusions as they do not have to make inferences from other (statistical) downscaling methods. The cyclone composites also clearly show the increase in wind speed and lower precipitation associated with these systems and therefore the authors reach the logical conclusion that the increased cyclone activity, combined with stronger low-level winds and lower precipitation, are likely to be responsible for the frequent dust storms the proxy data suggest. The manuscript is well written, provides a logical sequence of evidence and draws sound conclusions from them. The paper would be a good addition to the literature and should encourage other similar analyses for other models/epochs. I do have a fairly large number of minor points that should be considered, but they should be very straightforward to address.

SPECIFIC POINTS

Line 19/20: "...which is typically..." change to "...which are typically..."

End of line 37: change stronger to larger ("larger differences" reads better).

Lines 41-42: Re-word to something more concise like, "One important issue preventing non-recent or non-21st century cyclone analysis is the availability of climate model output with sufficient spatial and temporal resolution to enable identification, tracking and characterisation of such cyclones."

Line 56: Change "enhanced" to "stronger".

Lines 57-58: change to: "...leading to a southward displaced, more intense and less variable North Atlantic jet than under..."

Line 59: change "related with" to "related to".

Lines 68-71: the sentence starting "For example," in line 68 is too long and I struggled to make sense of it. Please would you break it up into two sentences and re-word it.

Line 66: insert "the" before "LGM in PMIP3". Whole paragraph, lines 56-74: There is this paper (below) that suggests reduced storminess over the North Atlantic at the LGM, which the authors should consider as a counter-view to their paper. It does not invalidate the results here whatsoever, but does provide an important (and perhaps opposing) view of the North Atlantic at the LGM. Just a sentence acknowledging this and citing the paper would be sufficient: Rivière, G., S. Berthou, G. Lapeyre, and M. Kageyama, 2018: On the Reduced North Atlantic Storminess during the Last Glacial Period: The Role of Topography in Shaping Synoptic Eddies. J. Climate, 31, 1637–1652, https://doi.org/10.1175/JCLI-D-17-0247.1

Line 78: change to "...compared to their modern counterparts at high spatial...".

Line 84: change to "...our analysis is data from the third phase..."

Lines 87-88: change "adapted" to "lower" as the greenhouse gas concentrations are lower.

Line 90: Remove the "have" after the Ludwig et al. (2016) reference.

Line 91: Include "the" before "LGM".

Line 92: Change to "...slightly different jet structure to some of the other..."

Lines 92-93: remove the words, "in terms of the difference between PI and LGM conditions" as they unnecessarily make the sentence too long.

Lines 108-109: change to "...to simulate the TOP 30 cyclones (from PI and LGM) with a grid spacing of..."

Lines 112-114: Different spellings of parametrisation/parametrization – make sure you are consistent (either way seems OK with Copernicus Publications but you need to be consistent – see manuscript preparation guidelines).

Line 112: Change to "An overview of the parametrisation choices is given in Table 2." As it is more concise. Paragraph for lines 119-126: Start the paragraph with something like, "The TOP 30 cyclone tracks simulated by WRF were identified manually...". Just to be clear that you have not manually tracked all cyclones in these simulations.

Line 121: Change "equally" to "each".

Line 123: Change "For the sake of succinctness," to "For brevity,".

Line 124: Change "The here analysed target variables based on the 12.5..." to "The variables analysed from the 12.5...".

Lines 130-131: I think you should actually include the plot of MPI-ESM-P vs reanalyses in the paper to help this section. It makes it quicker and easier for the reader to verify your statement. Including the figure in the supplement would be absolutely fine.

Lines 136-137: Change "The North Atlantic storm track looks quite different under LGM conditions: the cyclone track density is strongly enhanced over the North Atlantic and more constraint along a corridor close to the ice edge (Fig. 2b)." to "The North Atlantic storm track looks quite different under LGM conditions relative to PI: the cyclone track density is higher over the North Atlantic and more constrained to the ice edge (Fig. 2b)."

Line 142: I read the numbers 12.071 vs. 9.541 as "twelve versus nine-and-a-half" cyclones over a 30 year period. Is this supposed to be 1000 times that i.e. twelve and nine thousand, respectively over a 30 year period? That would be about 2.2/day vs. 1.8/day, which seems right. If it is 'thousands' of cyclones then just put 12071 and 9541 as the inclusion of the decimal point could mean different things to different readers and removing it would remove any confusion.

Line 144: add 'the' here, "for the LGM"

Line 146: change "cyclogenesis is enhanced" to "the rate of cyclogenesis has increased". I find the use of "enhanced" can be ambiguous. Do you mean the cyclones

deepen faster? More cyclones generated? Other features associated with cyclogenesis strengthen? Stating that the number of storms being formed has increased (instead of enhanced) makes the sentence much clearer. Please consider the use of "enhanced" elsewhere and whether a more direct statement can be made (as above).

Line 146: regarding enhanced. . . you could say, "On the other hand, there is more cyclolysis along the borders. . .". Again, what aspect of cyclolysis is "enhanced"? If you mean more storms are decaying there then you can just use the suggested wording above.

Line 147: Change "in strong deviation" to "relative".

Line 149: remove the word "reveal" and replace with "have".

Line 152: remove "particularly".

Line 153: do you mean ice sheets not shields?

Lines 153-154: Sentence starting "This is in strong. . ." does not seem to quite make sense to me and I am not sure how to re-word it. Could you check to make sure it is clearly stating what you want it to say.

Line 158: "LGM cyclones are on average more intense than their PI counterparts". While I accept this is true, your analysis/summary of the figure does not clearly show this. All I can see is a skewed distribution. You just need to quantify this and then quote the mean (and median, given the distribution shape) in the text to back this statement up.

Line 169: Again, quote the actual mean/median value from the cyclones in Table 3 to clearly show that the vorticity is higher for the LGM relative to PI.

Line 188: Change "Take" to "Taking".

Line 193: Change to ". . .temperature at 850 hPa for LGM relative to PI".

Line 194: Change "apparently faster" to "displaced forward in the cyclone".

Line 200: Change "for LGM extremely cyclones" to "for the LGM extreme cyclones"

Line 202: Change "strongly increased" to "much higher".

Line 203: Change m/s to m s-1.

Line 213: Change "occurring climate" to "climatic conditions".

Line 214: Change "under LGM conditions" to "at the LGM".

Line 229: Just say "shortcomings" not "some shortcomings", also give an example the sort of shortcoming you are referring to.

Line 230: Change "this caveat" to "those shortcomings".

Line 231: Change "Pfahl et al. (2015) had" to "Pfahl et al. (2015) has".

Lines 234-239: Sentence starting "Even though…" is far too long and needs to be broken into at least two sentences.

Line 240: change to "close to the Iberian Peninsula for cyclone selection leads…".

Line 243: add 'the' – "typically initiated by the wind…".

Line 246: change to "have been stronger than at present… (Maher…)".

Line 247: Change "document of a past when" to "indicate that"

Line 250: What's an "adequate large-scale circulation"? Do you mean, "stronger large-scale flow"?

Line 255: remove "European" as you state Europe at the start of the sentence.

Line 257-258: change to "…individual LGM cyclones could trigger such dust…".

Line 258: Change "strong wind speeds" to just "strong winds".

Line 259: change to ". . .trigger dust emission and transport over short. . .".

Line 260-261: sentence starting "As the precipitation. . ." was confusing. You need to say something like, "As moisture acts to make surface dust particles more cohesive (REFERENCE*), the reduced cyclone precipitation and higher wind speeds in LGM cyclones would have actually been more conducive to generating dust storms" – or something close to those words. *please find and insert an appropriate reference here.

Line 273: change "featuring an enhanced number and" to "featuring more frequent and intense" which is much clearer.

Lines 276-277: Change "LGM cyclones were more intense due to stronger baroclinicity and apparently less influenced by diabatic processes (lower rainfall, lower temperatures, lower water vapour content)." to "LGM cyclones were more intense due to stronger baroclinicity with less influence from diabatic processes (lower rainfall and lower water vapour content)." Note: I removed 'lower temperature' as temperature can change adiabatically, which matters given e.g. lower sea levels and therefore higher mean sea level pressure, which would affect temperature.

Line 293: change "in favour of an extended used" to "for the extended use".

Figure 1 caption: change "extends" to "extents".

Figure 3 caption: change ". . .ice sheet extends marked by blue line. . ." to "ice sheet extents marked by the blue line. . .".

Figure 4 caption: change ". . . the y-axis is zoomed in." to ". . .the y-axis is adjusted (right figure)."

Figure 6: the MSLP in WRF 12.5 Fig. 6(d) is lower at the cyclone centre than WRF 50's cyclone centre in Fig. 6(c) but the MSLP is higher (and not even below 990 hPa, whereas I can see a 985 hPa closed contour in Fig. 6d) in the WRF 12.5 km than either of the other two models. This does not look correct. Please check this as the MSLP changes in Fig 6(b) would not be consistent with the vorticity changes given the

method of calculating vorticity.

Figure 7: The caption is incorrect. Where you refer to (d, e) and (f – g) I think you mean (d – f) and (g – i). Please re-read the caption carefully and make sure it corresponds to the correct figure panels. Also, you refer to the 'field mean' in the caption – averaged over what area? The cyclone area? Hemisphere? Globe? Please be clear on that.

Figure S1: The same issue as described for the Figure 7 caption applies to this figure too. Please check through it.

———————————————

---

## Referee Comment (RC2) · Anonymous Referee #2 · 6 Jan 2020

This is an interesting paper assessing how cyclone activity differed in parts of the northern hemisphere during the last glacial maximum. The paper is well written and the results are interesting, particularly the cyclone composites from the regional model simulations showing the different cyclone impacts between the two periods.

My major comment is that the ice sheets and sea level change resulted in quite substantial differences in topography, MSLP etc between the two periods, which influences how the tracking scheme functions. I think there would be a great deal of value in using a higher level in the atmosphere (e.g. 500hPa) for identifying cyclones to assess

how robust the observed changes are. I recognise that this is difficult (due to the time involved) or possibly impossible, depending on what data exists for the global model, but it could provide some useful insights, even just from the WRF simulations.

Minor comments:

L89 - I would appreciate more information on why this specific global model was chosen, especially as you say its large-scale circulation is different from other models. Please also note its spatial resolution.

L119 - does identifying cyclones manually make much difference compared to the tracking scheme used for the GCM cyclones?

L130 - I would appreciate some more evaluation of how the model compares to reanalyses included within this paper.

L173 - please elaborate on how "care was taken" that the tracks align. It would be nice to see some statistics comparing e.g. mean biases in location/intensity, rather than just for a selected cyclone.

Table 3 - In addition (or instead of) these stats for all cyclones, some summary statistics of the top 30 cyclones for each simulation would be helpful. E.g. mean intensity and cyclogenesis latitude from the GCM, or mean wind speed, rain rate, etc from the WRF simulations.

Figure 6 c/d - it would be nice to have an additional panel showing the MSLP field in the GCM

Figure 7 - are the differences in panels c and f between the absolute values or between the anomalies? Given that there is the large mean MSLP difference between the simulations, I think the difference in local anomalies might be more informative for understanding how e.g. pressure gradients differ between simulations.

---

## Short Comment (SC1) · 21 Jan 2020

I really enjoyed reading this manuscript and I hope the review process will be smooth. I have two small comments/requests though:

(1) This manuscript was recently accepted for publication in EPSL. Among other things, it presents a new interpretation of the LGM jet zonalization in the N Atlantic and the precipitation distribution in SW Europe. I would greatly appreciate if the authors could cite this paper when you discuss these topics in the introduction.  Link to accepted

manuscript:

https://authors.elsevier.com/a/1aRAS,Ig4KpRO

(2) The main result of the present study is at odds with a number of GCM studies that, contrary to findings here, show a reduced storminess in the N Atlantic at the LGM. It would be good to add a paragraph in the discussion section that puts these contrasting results in perspective with one another, and if possible, provide at least a speculative interpretation of possible sources of this discrepancy. For example, it could be model dependent (the top three studies used CCSM3 derivatives, and Riviere et al used IPSL), resolution could be a factor (this paper discussed this aspect in some detail in GCM simulations – see e.g. Fig 1: https://www.the-cryosphere.net/12/1499/2018/tc-12-1499-2018.html), parameterizations, boundary conditions (again, top three studies used PMIP2 boundary conds.), etc.

Li and Battisti, 2008 JClim https://journals.ametsoc.org/doi/full/10.1175/2007JCLI2166.1

Donohoe and Battisti, 2009 JClim https://journals.ametsoc.org/doi/full/10.1175/2009JCLI2776.1

Lofverstrom et al. 2016 JAS https://journals.ametsoc.org/doi/full/10.1175/JAS-D-15-0295.1

Riviere et al, 2018, JClim https://journals.ametsoc.org/doi/full/10.1175/JCLI-D-17-0247.1

---

## Author Comment (AC1) · 12 Feb 2020

Replies to Short Comment #1

Marcus Lofverstrom (lofverstrom@arizona.edu)

Original comments received and published: 6 January 2020

I really enjoyed reading this manuscript and I hope the review process will be smooth. I have two small comments/requests though:

Answer: We want to thank the Colleague for his comments to our manuscript. We reply point-by-point to the Reviewer's comments. Our responses are shown in red color. Text from the manuscript is identified by quotation marks and italic font style, added or modified text can be identified by red colour.

(1) This manuscript was recently accepted for publication in EPSL. Among other things, it presents a new interpretation of the LGM jet zonalization in the N Atlantic and the precipitation distribution in SW Europe. I would greatly appreciate if the authors could cite this paper when you discuss these topics in the introduction. Link to accepted manuscript:

https://authors.elsevier.com/a/1aRAS,Ig4KpRO

A: We thank the colleague for this suggestion. We now included a reference to this paper in the introduction (line 60, see text in next reply).

(2) The main result of the present study is at odds with a number of GCM studies that, contrary to findings here, show a reduced storminess in the N Atlantic at the LGM. It would be good to add a paragraph in the discussion section that puts these contrasting results in perspective with one another, and if possible, provide at least a speculative interpretation of possible sources of this discrepancy. For example, it could be model dependent (the top three studies used CCSM3 derivatives, and Riviere et al used IPSL), resolution could be a factor (this paper discussed this aspect in some detail in GCM simulations – see e.g. Fig 1: https://www.the-cryosphere.net/12/1499/2018/tc-12-1499-2018.html), parameterizations, boundary conditions (again, top three studies used PMIP2 boundary conds.), etc.

Li and Battisti, 2008 JClim https://journals.ametsoc.org/doi/full/10.1175/2007JCLI2166.1

Donohoe and Battisti, 2009 JClim
https://journals.ametsoc.org/doi/full/10.1175/2009JCLI2776.1

Lofverstrom et al. 2016 JAS https://journals.ametsoc.org/doi/full/10.1175/JAS-D-15-0295.1

Riviere et al, 2018, JClim https://journals.ametsoc.org/doi/full/10.1175/JCLI-D-17-0247.1

A: We also thank the colleague for this suggestion. A similar comment had been posted by reviewer #1. We have enhanced our discussion on this issue and explicitly state that the changes in storminess in the PMIP3 models depends on the model choice, parameterizations and chosen boundary conditions.

*"Under the influence of the continental ice sheets and extended sea ice, the PMIP3 GCMs show stronger meridional temperature gradients, leading to a southward displaced, more intense and less variable North Atlantic jet than under current climate conditions (Löfverström et al, 2014; 2016; Merz et al., 2015; Wang et al. 2018). These differences have been related e.g., to more dominant cyclonic Rossby wave breaking near Greenland (Riviére et al., 2010), stationary wave packets trapped in the mid-latitude wave guide (Löfverström, 2020) and to enhanced meridional eddy momentum flux convergence over the North Atlantic (Wang et al., 2018). In line with a southward displaced and stronger jet stream, several studies show a more intense and southward shifted North Atlantic storm track compared to today's climate (e.g., Hofer et al., 2012; Luetscher et al., 2015; Ludwig et al., 2016). However, other studies display reduced storm track activity over the North Atlantic in spite of the enhanced baroclinicity (e.g. Donohoe and Battisti, 2009; Riviére et al., 2010; Löfverström et al., 2016). Riviére et al. (2018) discusses a reduced baroclinic conversion as a possible reason for this apparent discrepancy, arguing that the eddy heat fluxes are less well aligned with the mean temperature gradient for LGM than for PI. Other arguments for the reduced storminess include model resolution, parameterizations and boundary conditions (e.g. Donohoe and Battisti, 2009; Riviére et al., 2018). Thus, the intensity differences between LGM and PI North Atlantic storm track activity may be model dependent."* (lines 56ff).

In the discussion, we have added:

*"Given that this study is based on a single GCM, a single tracking method and a single RCM, it should be regarded as a preliminary analysis as the uncertainties of the jet stream position and storm track activity (e.g., Merz et al, 2013, Riviere et al., 2018) may be considerable among different GCMs."* (lines 293ff)

---

## Author Comment (AC2) · 12 Feb 2020

Replies to Anonymous Referee #1

Original comments received and published: 16 December 2019

GENERAL COMMENTS

This paper outlines the use of cyclone tracking to identify some key points of the climate and weather over the North Atlantic and Western Europe during that Last Glacial Maximum (LGM). The paper identifies changes in the mean state that may be both conducive to dust emission (e.g. increase in wind speeds) and also those which may hinder it (e.g. more storm activity and associated precipitation). The key point in the authors' analysis is the cyclone tracking and compositing, which provides important information on how the mean climate state is felt at the surface through the extratropical cyclones (in this case, the 30 most extreme). It is very clear from their results that the strengthening of the thermal gradients in the LGM (relative to the base period) simulations do not cause an increase in precipitation (at the fronts) and that overall there is less precipitation in the extratropical cyclones (for clear physical reasons, which are stated). The high-resolution model data is also useful for reaching their conclusions as they do not have to make inferences from other (statistical) downscaling methods.

The cyclone composites also clearly show the increase in wind speed and lower precipitation associated with these systems and therefore the authors reach the logical conclusion that the increased cyclone activity, combined with stronger low-level winds and lower precipitation, are likely to be responsible for the frequent dust storms the proxy data suggest. The manuscript is well written, provides a logical sequence of evidence and draws sound conclusions from them. The paper would be a good addition to the literature and should encourage other similar analyses for other models/epochs. I do have a fairly large number of minor points that should be considered, but they should be very straightforward to address.

Answer: We want to thank the Reviewer for the thorough examination and positive assessment of our manuscript. We reply point-by-point to the Reviewer's comments or give detailed arguments about our reasoning for those cases in which we did not follow a Reviewer suggestion. Our responses are shown in red colour. Text from the manuscript is identified by quotation marks and italic font style, added or modified text can be identified by colour.

SPECIFIC POINTS

Important note: In case no specific answer can be found below a comment, we have simply implemented the suggested text changes as requested.

Line 19/20: ". . .which is typically. . ." change to ". . .which are typically. . ."

End of line 37: change stronger to larger ("larger differences" reads better).

Lines 41-42: Re-word to something more concise like, "One important issue preventing non-recent or non-21st century cyclone analysis is the availability of climate model output with sufficient spatial and temporal resolution to enable identification, tracking and characterisation of such cyclones."

Line 56: Change "enhanced" to "stronger".

Lines 57-58: change to: ". . .leading to a southward displaced, more intense and less variable North Atlantic jet than under. . ."

Line 59: change "related with" to "related to".

Lines 68-71: the sentence starting "For example," in line 68 is too long and I struggled to make sense of it. Please would you break it up into two sentences and re-word it.

A: we have changed the sentence as follows:

*"For example, Ludwig et al (2017) implemented more realistic boundary conditions in terms of the North Atlantic SSTs, land use types and vegetation cover in a regional climate model (RCM) to simulate the regional climate under LGM conditions. Their results in terms of LGM temperature, precipitation and the permafrost margin are in better agreement with the proxies than without the implemented boundary conditions." (line 74ff)*

Line 66: insert "the" before "LGM in PMIP3".

Whole paragraph, lines 56-74: There is this paper (below) that suggests reduced storminess over the North Atlantic at the LGM, which the authors should consider as a counter-view to their paper. It does not invalidate the results here whatsoever, but does provide an important (and perhaps opposing) view of the North Atlantic at the LGM. Just a sentence acknowledging this and citing the paper would be sufficient: Rivière, G., S. Berthou, G. Lapeyre, and M. Kageyama, 2018: On the Reduced North Atlantic Storminess during the Last Glacial Period: The Role of Topography in Shaping Synoptic Eddies. J. Climate, 31, 1637– 1652, https://doi.org/10.1175/JCLI-D-17-0247.1

A: we thank the reviewer for his comment and the reference, which we have implemented in the text and shortly discussed. Additional comments from M. Löfverström (SC1) were also included.

*"Under the influence of the continental ice sheets and extended sea ice, the PMIP3 GCMs show stronger meridional temperature gradients, leading to a southward displaced, more intense and less variable North Atlantic jet than under current climate conditions (Löfverström et al, 2014; 2016; Merz et al., 2015; Wang et al. 2018). These differences have been related e.g., to more dominant cyclonic Rossby wave breaking near Greenland (Riviére et al., 2010), stationary wave packets trapped in the mid-latitude wave guide (Löfverström, 2020) and to enhanced meridional eddy momentum flux convergence over the North Atlantic (Wang et al., 2018). In line with a southward displaced and stronger jet stream, several studies show a more intense and southward shifted North Atlantic storm track compared to today's climate (e.g., Hofer et al., 2012; Luetscher et al., 2015; Ludwig et al., 2016). However, other studies display reduced storm track activity over the North Atlantic in spite of the enhanced baroclinicity (e.g. Donohoe and Battisti, 2009; Riviéré et al., 2010; Löfverström et al., 2016). Riviéré et al. (2018) discusses a reduced baroclinic conversion as a possible reason for this apparent discrepancy, arguing that the eddy heat fluxes are less well aligned with the mean temperature gradient for LGM than for PI. Other arguments for the reduced storminess include model resolution, parameterizations and boundary conditions (e.g. Donohoe and Battisti, 2009; Riviéré et al., 2018). Thus, the intensity differences between LGM and PI North Atlantic storm track activity may be model dependent." (lines 56ff).*

Line 78: change to ". . .compared to their modern counterparts at high spatial. . .".

Line 84: change to ". . .our analysis is data from the third phase. . ."

Lines 87-88: change "adapted" to "lower" as the greenhouse gas concentrations are lower.

Line 90: Remove the "have" after the Ludwig et al. (2016) reference.

Line 91: Include "the" before "LGM".

Line 92: Change to ". . .slightly different jet structure to some of the other. . ."

Lines 92-93: remove the words, "in terms of the difference between PI and LGM conditions" as they unnecessarily make the sentence too long.

Lines 108-109: change to ". . .to simulate the TOP 30 cyclones (from PI and LGM) with a grid spacing of. . ."

Lines 112-114: Different spellings of parametrisation/parametrization – make sure you are consistent (either way seems OK with Copernicus Publications but you need to be consistent – see manuscript preparation guidelines).

Line 112: Change to "An overview of the parametrisation choices is given in Table 2." As it is more concise.

Paragraph for lines 119-126: Start the paragraph with something like, "The TOP 30 cyclone tracks simulated by WRF were identified manually. . .". Just to be clear that you have not manually tracked all cyclones in these simulations.

Line 121: Change "equally" to "each".

Line 123: Change "For the sake of succinctness," to "For brevity,".

Line 124: Change "The here analysed target variables based on the 12.5. . ." to "The variables analysed from the 12.5. . .".

Lines 130-131: I think you should actually include the plot of MPI-ESM-P vs reanalyses in the paper to help this section. It makes it quicker and easier for the reader to verify your statement. Including the figure in the supplement would be absolutely fine.

A: We thank the reviewer for this suggestion. As requested, we have added a further figure to the supplementary material showing the difference between MPI-ESM-P (PI and LGM) and the NCEP/NCAR reanalyses (Figure S1 +S2) and added a short notice at the beginning of chapter 3.

[Figure]

Fig. S1: Top: storm tracks (2–6 days band passed filter of daily MSLP data [1/10 hPa]) for the NCEP Reanalysis data and the MPI-ESM-P simulations for PI and LGM. Bottom: differences (shaded) between PI (lines) and NCEP and between LGM (lines) and PI. Areas with topography higher 1000 m shaded grey, LGM ice sheet extent marked by the blue line.

[Figure]

Fig. S2: Top: Upper level jet stream (wind speed at 300 hPa [m/s]) for the NCEP Reanalysis data and the MPI-ESM-P simulations for PI and LGM. Bottom: differences (shaded) between PI (lines) and NCEP and between LGM (lines) and PI. LGM ice sheet extent marked by the blue line.

Lines 136-137: Change "The North Atlantic storm track looks quite different under LGM conditions: the cyclone track density is strongly enhanced over the North Atlantic and more constraint along a corridor close to the ice edge (Fig. 2b)." to "The North Atlantic storm track looks quite different under LGM conditions relative to PI: the cyclone track density is higher over the North Atlantic and more constrained to the ice edge (Fig. 2b)."

Line 142: I read the numbers 12.071 vs. 9.541 as "twelve versus nine-and-a-half" cyclones over a 30 year period. Is this supposed to be 1000 times that i.e. twelve and nine thousand, respectively over a 30 year period? That would be about 2.2/day vs. 1.8/day, which seems right. If it is 'thousands' of cyclones then just put 12071 and 9541 as the inclusion of the decimal point could mean different things to different readers and removing it would remove any confusion.

A: we agree the text was not very clear, and has been changed as follows:

"For the North Atlantic (70°W – 0°, 35°N – 70°N), the total number of cyclones for the analysed 30-year period is about 26% larger for LGM than for PI conditions (12071 vs. 9541 individual cyclone counts in 30 years, corresponding to roughly 2.2 cyclones/day vs 1.8 cyclones/day)." (lines 148ff)

Line 144: add 'the' here, "for the LGM"

Line 146: change "cyclogenesis is enhanced" to "the rate of cyclogenesis has increased". I find the use of "enhanced" can be ambiguous. Do you mean the cyclones deepen faster? More cyclones generated? Other features associated with cyclogenesis strengthen? Stating that the number of storms being formed has increased (instead of enhanced) makes the sentence much clearer. Please consider the use of "enhanced" elsewhere and whether a more direct statement can be made (as above).

Line 146: regarding enhanced. . . you could say, "On the other hand, there is more cyclolysis along the borders. . .". Again, what aspect of cyclolysis is "enhanced"? If you mean more storms are decaying there then you can just use the suggested wording above.

Line 147: Change "in strong deviation" to "relative".

Line 149: remove the word "reveal" and replace with "have".

Line 152: remove "particularly".

Line 153: do you mean ice sheets not shields?

Lines 153-154: Sentence starting "This is in strong. . ." does not seem to quite make sense to me and I am not sure how to re-word it. Could you check to make sure it is clearly stating what you want it to say.

A: We thank the reviewer for pointing this out, we agree that the wording was not clear, and that this sentence kind of duplicates what is said in the previous sentence. We have deleted the sentence, and slightly changed the previous sentence.

*"Deepening rates are stronger for LGM cyclones over the central north Atlantic, as well as their filling rates close to the ice edge / ice sheets (Fig. 3g,h)." (lines 164ff)*

Line 158: "LGM cyclones are on average more intense than their PI counterparts". While I accept this is true, your analysis/summary of the figure does not clearly show this. All I can see is a skewed distribution. You just need to quantify this and then quote the mean (and median, given the distribution shape) in the text to back this statement up.

A: Thanks for this, we have added the numbers to the text.

*"Figure 4 displays the relative frequency distribution of the cyclone intensity over the North Atlantic area (70°W – 0°, 35°N – 70°N), revealing that LGM cyclones are on average more intense (mean (median): 1.58 (1.41) hPa deg. lat.$^{-2}$s$^{-1}$) than their PI counterparts (mean (median): 1.43 (1.28) hPa deg. lat.$^{-2}$s$^{-1}$)." (lines 168ff)*

Line 169: Again, quote the actual mean/median value from the cyclones in Table 3 to clearly show that the vorticity is higher for the LGM relative to PI.

A: Again, we added the numbers to the text (see above). Additionally, we added a line to the table including mean/median values for better comparison.

*"LGM extreme cyclone trajectories are more zonally orientated and constrained to a narrower corridor (particularly until 15°W) than their PI counterparts, and they achieve higher vorticity (mean Laplacian of MSLP for LGM: 2.80; PI: 2.52) values during lifetime (Tab. 3)." (lines 178ff)*

Line 188: Change "Take" to "Taking".

Line 193: Change to ". . .temperature at 850 hPa for LGM relative to PI".

Line 194: Change "apparently faster" to "displaced forward in the cyclone".

Line 200: Change "for LGM extremely cyclones" to "for the LGM extreme cyclones"

Line 202: Change "strongly increased" to "much higher".

Line 203: Change m/s to m s-1.

Line 213: Change "occurring climate" to "climatic conditions".

Line 214: Change "under LGM conditions" to "at the LGM".

Line 229: Just say "shortcomings" not "some shortcomings", also give an example the sort of shortcoming you are referring to.

Line 230: Change "this caveat" to "those shortcomings".

Line 231: Change "Pfahl et al. (2015) had" to "Pfahl et al. (2015) has".

Lines 234-239: Sentence starting "Even though. . ." is far too long and needs to be broken into at least two sentences.

A: We agree the sentence was too long and have broken it into two separate sentences

*"Even though lower cyclone-related precipitation may be partially compensated by the moisture advection embedded in the (stronger) westerly large-scale flow, particularly for areas where orographic precipitation dominates (e.g. upward slopes of mountain ranges / glaciers), it is consistent with the hypothesis of a drier Western and Central Europe. The view of a drier Europe is also consistent with the dominant land cover types estimated from proxy data (Ray and Adams, 2001) or a statistical reconstruction based on temperature and precipitation (Shao et al., 2018), namely polar desert close to the glaciers, forest steppe over Southern Europe and steppe in-between." (line 243ff)*

Line 240: change to "close to the Iberian Peninsula for cyclone selection leads. . .".

Line 243: add 'the' – "typically initiated by the wind. . .".

Line 246: change to "have been stronger than at present. . . (Maher. . .)".

Line 247: Change "document of a past when" to "indicate that"

Line 250: What's an "adequate large-scale circulation"? Do you mean, "stronger largescale flow"?

A: Yes, this is what we meant, text changed (line 259)

Line 255: remove "European" as you state Europe at the start of the sentence.

Line 257-258: change to ". . .individual LGM cyclones could trigger such dust. . .".

Line 258: Change "strong wind speeds" to just "strong winds".

Line 259: change to ". . .trigger dust emission and transport over short. . .".

Line 260-261: sentence starting "As the precipitation. . ." was confusing. You need to say something like, "As moisture acts to make surface dust particles more cohesive (REFERENCE*), the reduced cyclone precipitation and higher wind speeds in LGM cyclones would have actually been more conducive to generating dust storms" – or something close to those words. *please find and insert an appropriate reference here.

A. Thanks for the suggestion, implemented as suggested:

*"As moisture acts to make surface dust particles more cohesive (e.g., Ishizuka et al., 2008), the reduced cyclone precipitation and higher wind speeds in LGM cyclones would have actually been more conducive to generating dust storms." (lines 268ff)*

Line 273: change "featuring an enhanced number and" to "featuring more frequent and intense" which is much clearer.

Lines 276-277: Change "LGM cyclones were more intense due to stronger baroclinicity and apparently less influenced by diabatic processes (lower rainfall, lower temperatures, lower water vapour content)." to "LGM cyclones were more intense due to stronger baroclinicity with less influence from diabatic processes (lower rainfall and lower water vapour content)." Note: I removed 'lower temperature' as temperature can change adiabatically, which matters given e.g. lower sea levels and therefore higher mean sea level pressure, which would affect temperature.

Line 293: change "in favour of an extended used" to "for the extended use".

Figure 1 caption: change "extends" to "extents".

Figure 3 caption: change ". . .ice sheet extends marked by blue line. . ." to "ice sheet extents marked by the blue line. . .".

Figure 4 caption: change ". . . the y-axis is zoomed in." to ". . .the y-axis is adjusted (right figure)."

Figure 6: the MSLP in WRF 12.5 Fig. 6(d) is lower at the cyclone centre than WRF 50's cyclone centre in Fig. 6(c) but the MSLP is higher (and not even below 990 hPa, whereas I can see a 985 hPa closed contour in Fig. 6d) in the WRF 12.5 km than either of the other two models. This does not look correct. Please check this as the MSLP changes in Fig 6(b) would not be consistent with the vorticity changes given the method of calculating vorticity.

A: Thanks for pointing this out. There were wrong settings to the axes handling in the plotting script which has been corrected. The MSLP changes are now consistent with the changes in vorticity. See updated figure below:

[Figure]

*"Figure 6. Comparison of (a) cyclone tracks for MPI-ESM (black), WRF 50km (red) and WRF 12.5 km (blue) (coloured dots mark the location of peak intensity, black dotted box shows target area) and MSLP [hPa] for MPI-ESM at peak intensity, (b) timeseries of cyclone core pressure and relative vorticity for MPI-ESM, WRF 50km and WRF 12.5 km for LGM cyclone #24. Simulated precipitation rate [mm h-1] (shaded) and MSLP [hPa] (lines) at peak intensity for (c) WRF 50km and (d) WRF 12.5km."*

Figure 7: The caption is incorrect. Where you refer to (d, e) and (f – g) I think you mean (d – f) and (g – i). Please re-read the caption carefully and make sure it corresponds to the correct figure panels. Also, you refer to the 'field mean' in the caption – averaged over what area? The cyclone area? Hemisphere? Globe? Please be clear on that.

Figure S1: The same issue as described for the Figure 7 caption applies to this figure too. Please check through it.

A: We have reworded the captions in both Fig. 7 and S3 (former Fig. S1) and added information about the field mean (which is the mean over the displayed area).

*"Figure 7. Composites of (a - c) mean sea level pressure, (d - f) ThetaE, and (g - i) vertical integrated water vapour (IWV) for PI, LGM and difference LGM – PI at peak intensity as defined by the maximum of the Laplacian of MSLP. (a, b) absolute MSLP values (lines, [hPa]), anomalies [hPa] from mean over displayed area (coloured); (c) absolute MSLP values (lines, [hPa]) from LGM, differences of the anomalies between LGM – PI in colours; (d, e) absolute ThetaE values (lines, [K]) and anomalies [K] from mean over displayed area (coloured); (f) absolute ThetaE values (lines, [K]) from LGM, differences of the anomalies between LGM – PI in colours; (g, h) absolute IWV values [mm], (i) difference [mm] LGM – PI."*

---

## Author Comment (AC3) · 12 Feb 2020

Replies to Anonymous Referee #2

Original comments received and published: 6 January 2020

This is an interesting paper assessing how cyclone activity differed in parts of the northern hemisphere during the last glacial maximum. The paper is well written and the results are interesting, particularly the cyclone composites from the regional model simulations showing the different cyclone impacts between the two periods.

Answer: We want to thank the Reviewer and the Editor for the thorough examination and positive assessment of our manuscript. We reply point-by-point to the Reviewer's comments or give detailed arguments about our reasoning for those cases in which we did not follow a Reviewer suggestion. Our responses are shown in red color. Text from the manuscript is identified by quotation marks and italic font style, added or modified text can be identified by red color.

My major comment is that the ice sheets and sea level change resulted in quite substantial differences in topography, MSLP etc between the two periods, which influences how the tracking scheme functions. I think there would be a great deal of value in using a higher level in the atmosphere (e.g. 500hPa) for identifying cyclones to assess how robust the observed changes are. I recognise that this is difficult (due to the time involved) or possibly impossible, depending on what data exists for the global model, but it could provide some useful insights, even just from the WRF simulations.

A: we thank the reviewer for this suggestion. As the reviewer correctly assumes, it would be very time consuming and in our opinion little informative to perform the tracking of cyclones at 500 hPa within the scope of this manuscript. Cyclones are typically tracked based on surface or near-surface fields, e.g. MSLP or relative vorticity at 850 hPa geopotential height (see Neu et al., 2013 for a description and comparison of many methodologies). In fact, the 500 hPa geopotential height field does not typically feature closed pressure systems, but rather troughs and ridges. In line with the reviewers suggestions, we thought about prepaing an additional figure showing the 500 hPa storm track field for the reanalysis, PI and LGM conditions. The storm track shows the variance of the 500 hPa geopotential height within the synoptic scale and is this a measure of synoptic activity (Hoskins and Valdes, 1990). Unfortunately, no 6-hourly nor even daily pressure level data is available in the PMIP3 database. This is why we have estimated the storm track in Ludwig et al. (2016) based on the MSLP fields (here also used for the cyclone tracking, see new Fig. S1). While a conversion between the MPI-ESM-P data from model levels to pressure levels is possible in

principle, we have refrained from taking this path as we expect little added information compared to the MSLP storm track (Ludwig et al., 2016) and cyclone track information (here)

Fig. S1: Top: storm tracks (2–6 days band passed filter of daily MSLP data [1/10 hPa]) for the NCEP Reanalysis data and the MPI-ESM-P simulations for PI and LGM. Bottom: differences (shaded) between PI (lines) and NCEP and between LGM (lines) and PI. Areas with topography higher 1000 m shaded grey, LGM ice sheet extent marked by the blue line.

We agree with the reviewer that looking at the vertical structure of the cyclones (see Dacre et al., 2012; their Fig. 4 https://doi.org/10.1175/BAMS-D-11-00164.1) in WRF for LGM and PI conditions would be very interesting. While we think this does not fit well with the scope of the present study, we will surely look into this in the future.

**Minor comments:**

L89 - I would appreciate more information on why this specific global model was chosen, especially as you say its large-scale circulation is different from other models. Please also note its spatial resolution.

A: We are based in Germany and we had access via the DKRZ to the 6-hourly 3D MPI-ESM-P fields. These served as boundary conditions for the WRF simulations. Please note that the surface and atmospheric fields stored on the PMIP3 database are not sufficient to serve as boundary conditions for a regional climate model. Theoretically, we could ask each modelling group for the high resolution 3D-data and re-do the current analysis in case the needed data is available. This is one of the reasons we state in the conclusions that the analysis should be expanded based on other GCMs (and also RCMs) to confirm (or not) the current results. We now indicate in the text that the choice of GCM was largely motivated by the data availability.

"This choice was motivated by the availability of six hourly 3-D model level data needed for running the RCM." (line 97)

L119 - does identifying cyclones manually make much difference compared to the tracking scheme used for the GCM cyclones?

A: The tracking scheme applied to the GCM data is fully automatic (Murray and Simmonds, 1991; Pinto et al., 2005) and widely used in the community (see e.g. review paper Neu et al 2013). These tracking schemes are optimised to process enormous amounts of data but with a rather low spatial and time resolution (e.g. 6-hourly, T63-data). For the RCM data, there was no need to use such a tracking method, particularly if only a very small amount of cyclones is analysed (TOP 30 for LGM and PI) and the spatial and time resolutions was very high (12.5 km, hourly). Given that the performance of the tracking methods typically increases with higher spatial and particularly time resolution, we assume that the results using a comparative tracking method would be similar to our manual analysis. Following one of the suggestions by the first reviewer, we now clearly indicate that the manual tracking was only performed for 2x 30 cyclones.

"The TOP 30 cyclone tracks simulated by WRF were identified manually based on relative vorticity field at 850 hPa." (line 126)

L130 - I would appreciate some more evaluation of how the model compares to reanalyses included within this paper.

A: We thank the reviewer for this suggestion. Also following a comment by the first reviewer, we have added a notes to the first and second paragraph of chapter 3 and two additional supplementary figures (Fig. S1, S2) comparing the storm track and the upper level jet stream in the MPI-ESM-P model for PI and LGM conditions and the NCEP/NCAR reanalysis.

"In this section, we analyse the general characteristics of cyclones over the North Atlantic and Europe under LGM conditions and compare them to PI climate conditions. Figure 2 shows the cyclone track density for the extended winter for PI and LGM climate conditions. In spite of the lower spatial resolution of MPI-ESM-P, the cyclone track density for the PI is close to cyclone statistics obtained with Reanalysis datasets, with slight southerly shift of cyclonic activity (see Figure S1 for comparison with NCEP reanalysis data (Kalnay et al, 1996)), and CMIP GCMs for recent climate conditions (cp. Pinto et al., 2007; their Figure 1). Still, some regional shortcomings are identified, notably the limited cyclone activity over the Mediterranean basin. The North Atlantic storm track shows a clear tilt towards Northern Europe and the Arctic Ocean for PI, and its location and orientation are closely related with the eddy-driven jet stream (black contours in Fig. 2a) and the associated upper-air baroclinicity (Hoskins and Valdes, 1990; Pinto et al., 2009). A comparison of the jet stream between MPI-ESM-P PI and NCEP Reanalysis data shows a slight tilt towards Europe by the MPI model (Fig. S2), in line with the enhanced (reduced) southward (northward) cyclone activity (Fig. S1)"

**Figure S1 (see above) reply to major concern**